# Functional and evolutionary analysis of host Synaptogyrin-2 in porcine circovirus type 2 susceptibility

**Lianna R. Walker**[1,2], **Hiep L. Vu**[1,3], **Kristi L. Montooth**[2], **Daniel C. Ciobanu**[1,2,3]*

**1** Animal Science Department, University of Nebraska-Lincoln, Lincoln, Nebraska, United States of America, **2** School of Biological Sciences, University of Nebraska-Lincoln, Lincoln, Nebraska, United States of America, **3** Nebraska Center for Virology, University of Nebraska-Lincoln, Lincoln, Nebraska, United States of America

* dciobanu@unl.edu

**Data Availability Statement:** All the data generated by this study, including numerical data for all of our graphs and summary statistics, is available as supplementary information (S1 Table.xlsx). Public swine DNA sequences used in this study were

## Abstract

Mammalian evolution has been influenced by viruses for millions of years, leaving signatures of adaptive evolution within genes encoding for viral interacting proteins. Synaptogyrin-2 (*SYNGR2*) is a transmembrane protein implicated in promoting bacterial and viral infections. A genome-wide association study of pigs experimentally infected with porcine circovirus type 2b (PCV2b) uncovered a missense mutation (*SYNGR2 p.Arg63Cys*) associated with viral load. In this study, CRISPR/Cas9-mediated gene editing of the porcine kidney 15 (PK15, *wtSYNGR2*$^{+p.63Arg}$) cell line generated clones homozygous for the favorable *SYNGR2 p.63Cys* allele (*emSYNGR2*$^{+p.63Cys}$). Infection of edited clones resulted in decreased PCV2 replication compared to wildtype PK15 (P<0.05), with consistent effects across genetically distinct PCV2b and PCV2d isolates. Sequence analyses of wild and domestic pigs (n>700) revealed the favorable *SYNGR2 p.63Cys* allele is unique to domestic pigs and more predominant in European than Asian breeds. A haplotype defined by the *SYNGR2 p.63Cys* allele was likely derived from an ancestral haplotype nearly fixed within European (0.977) but absent from Asian wild boar. We hypothesize that the *SYNGR2 p.63Cys* allele arose post-domestication in ancestral European swine. Decreased genetic diversity in homozygotes for the *SYNGR2 p.63Cys* allele compared to *SYNGR2 p.63Arg*, corroborates a rapid increase in frequency of *SYGNR2 p.63Cys* via positive selection. Signatures of adaptive evolution across mammalian species were also identified within *SYNGR2* intraluminal loop domains, coinciding with the location of *SYNGR2 p.Arg63Cys*. Therefore, *SYNGR2* may reflect a novel component of the host-virus evolutionary arms race across mammals with *SYNGR2 p.Arg63Cys* representing a species-specific example of putative adaptive evolution.

## Author summary

Our research provides direct evidence of the functional role of a missense substitution in the host *SYNGR2* gene (*SYNGR2 p.Arg63Cys*) in the replication of porcine circovirus 2,

obtained from data files available within the European Nucleotide Archive (ENA) database (https://www.ebi.ac.uk/ena/browser/home) that were generated from samples corresponding to a single individuals with defined wild or domestic status (S1 Table). The species trees were also obtained from NCBI for each of the taxonomic groups (https://www.ncbi.nlm.nih.gov/Taxonomy/CommonTree/wwwcmt.cgi).

**Funding:** This project was supported by Agriculture and Food Research Initiative Competitive Grant no. 2019-05380 from the USDA National Institute of Food and Agriculture to DCC and HV. The funders had no role in study design, data collection and analysis, decision to publish, or preparation of the manuscript.

**Competing interests:** The authors have declared that no competing interests exist.

the smallest virus known to infect mammalian cells. DNA sequences and haplotype analyses of *SYNGR2* across domestic and wild pig populations suggests a likely origin and subsequent positive selection of the favorable *SYNGR2 p.63Cys* allele post-domestication in European swine. While *SYNGR2 p.Arg63Cys* represents a species-specific example of putative adaptive evolution, signatures of selection detected in *SYNGR2* across mammalian species suggested that this gene is a new component of the host-virus evolutionary arms race.

## Introduction

The host-virus evolutionary arms race embodies an age-old battle to maintain fitness across all domains of life [1]. Recent evidence indicates that majority of genetic adaptation in host genomes occurs within viral interacting proteins (VIPs), which can interact with viruses in a strain-specific manner or across multiple members of a viral family [2]. While our knowledge of VIPs, especially in humans, has grown in recent years, the identification of adaptive genetic variants affecting host resilience remains limited. Additionally, the influence that viruses have on host evolution cannot be characterized by a single adaptive event, as the ability of viruses to rapidly counter-evolve is thought to drive repeated instances of adaptive change in VIPs [1,3]. Therefore, it's not only important to identify adaptive host alleles, but to also understand their function and context within the host-virus evolutionary arms race. Due to the constant threat imposed by viral pathogens, there is substantial interest in elucidating the role of host genetics in disease susceptibility across diverse fields from agriculture to human health.

Porcine circovirus type 2 (PCV2) is the smallest known virus capable of infecting mammalian cells with a single-stranded circular DNA genome approximately 1.7Kb in length. The majority of PCV2 strains belong to one of three major subtypes: PCV2a, PCV2b, and PCV2d. In the early 2000's a global genotype shift from PCV2a to PCV2b corresponded with increased incidence and severity of porcine circovirus associated diseases (PCVAD). However, evidence of increased PCV2d predominance in recent years suggests another global genotype shift may be in progress [4]. While only a fraction of pigs develop clinical disease, PCV2 is prevalent in domestic and wild populations around the world and continues to pose a constant threat to animal welfare. Previously, our group uncovered a host missense variant (*p.Arg63Cys*) within the host Synaptogyrin-2 (*SYNGR2*) gene that was associated with PCV2b viral load and immune response following *in vivo* experimental challenge [5]. Infection of a CRISPR-Cas9 mediated *SYNGR2* knock-out porcine kidney 15 (PK15) cell line significantly reduced PCV2b replication beginning 24 hours post infection [5]. However, direct experimental evidence for the causality of the *SYNGR2 p.Arg63Cys* variant or consistency of its effect across PCV2 subtypes have yet to be provided.

*SYNGR2* is a transmembrane protein with reported functions in vesicle biogenesis and membrane trafficking and transport [6]. Despite no known function in host immunity, *SYNGR2* has been recently implicated in promoting both viral and bacterial infections. Specifically, a direct interaction was reported between *SYNGR2* and the non-structural protein of a tick-borne human RNA virus, severe fever with thrombocytopenia syndrome virus (SFTSV) [7]. This interaction was found to be crucial for the formation of intracytoplasmic inclusion bodies utilized as viral factories for SFTSV replication [7]. Additionally, *SYNGR2* was found necessary for internalization of the active CdtB subunit and induced toxicity following exposure to *Aggregatibacter actinomycetemcomitans* cytotoxin [8,9]. Together, these studies indicate that *SYNGR2* may represent a non-immune VIP common across pathogens and

mammalian hosts. Therefore, investigating the evolution of *SYNGR2* in domestic pigs as well as other mammalian lineages could provide valuable insight and a model of host adaptation to co-evolving pathogens.

In this study, we provide direct experimental evidence for the function of the *SYNGR2 p.Arg63Cys* variant in the early stages of infection using *in vitro* models with consistent effects observed across PCV2 subtypes. Evaluation of *SYNGR2* sequence data revealed a likely origin and subsequent positive selection of the *SYNGR2 p.63Cys* allele post-domestication in European swine. Corresponding with the location of the *SYNGR2 p.Arg63Cys* variant, analyses for signatures of selection across mammalian species highlighted the intraluminal loop domains as key regions of putative adaptive evolution within *SYNGR2*. Therefore, *SYNGR2* may reflect a novel component of the host-virus evolutionary arms race across mammals with *SYNGR2 p.Arg63Cys* representing a species-specific example of putative adaptive evolution.

## Results

### The *SYNGR2 p.63Cys* allele confers a significant reduction in PCV2b replication

The porcine kidney 15 (PK15) cell line is of epithelial-origin and homozygous for the unfavorable *SYNGR2 p.63Arg* allele associated with higher viral load following *in vivo* experimental PCV2b challenge [5]. Edited PK15 clones homozygous for the alternate *SYNGR2 p.63Cys* allele were generated via CRISPR-Cas9 gene editing using a single guide RNA and DNA template homologous to the non-targeting strand (Figs A and B in S1 Text). To directly compare the effect of the *SYNGR2 p.Arg63Cys* variant on PCV2 infection, a *SYNGR2* knock-out PK15 clone (*emSYNGR2^-del^*) [5], an edited PK15 clone homozygous for the *SYNGR2 p.63Cys* allele (*emSYNGR2^+p.63Cys^*), and the wildtype PK15 cell line (*wtSYNGR2^+p.63Arg^*) were simultaneously infected with a PCV2b isolate. The number of PCV2b genome copies was significantly less for the *emSYNGR2^-del^* clone compared to either cell line with predicted functional *SYNGR2* proteins (P<0.05–0.001; Fig 1). However, a significant reduction in viral replication was also observed for the *emSYNGR2^+p.63Cys^* clone compared to *wtSYNGR2^+p.63Arg^* at 24 and 48 hours post infection (hpi) in the cell fraction (P<0.05–0.005) and at 72 hpi in the supernatant fraction (P<0.01; Fig 1). On average, the number of PCV2b copies after 72 hours increased by only 26-fold in *emSYNGR2^+p.63Cys^* cells and 85-fold in the supernatant compared to 77-fold and 1,177-fold in *wtSYNGR2^+p.63Arg^* cells and supernatant, respectively (P<0.05).

### Consistent effect of *SYNGR2 p.Arg63Cys* variant between genetically distinct PCV2 isolates

To assess whether the effect of the *SYNGR2 p.Arg63Cys* variant is consistent across PCV2 strains, these same edited and wildtype cell lines were infected with another PCV2 isolate belonging to the PCV2d subtype. Nucleotide and predicted amino acid sequence comparison confirmed the PCV2d isolate is genetically distinct from the PCV2b isolate with 96.15% similarity across the genome. The first open reading frame (ORF1), encoding the viral Replicase protein, exhibited greater similarity between isolates at both the nucleotide (97.46%) and amino acid level (99.04%) than ORF2, which encodes the viral Capsid protein. Specifically, ORF2 was 93.73% and 93.99% similar between isolates at the nucleotide and amino acid levels, respectively. Alignment of the predicted Capsid amino acid sequence revealed characteristic subtype-specific substitutions for both the PCV2b and PCV2d isolates [10]. For instance, both isolates have residues at positions 59 (R/K), 206 (I), and 63 (R), which form a large surface on the head of the Capsid protein, predicted to increase host-cell binding affinity compared to

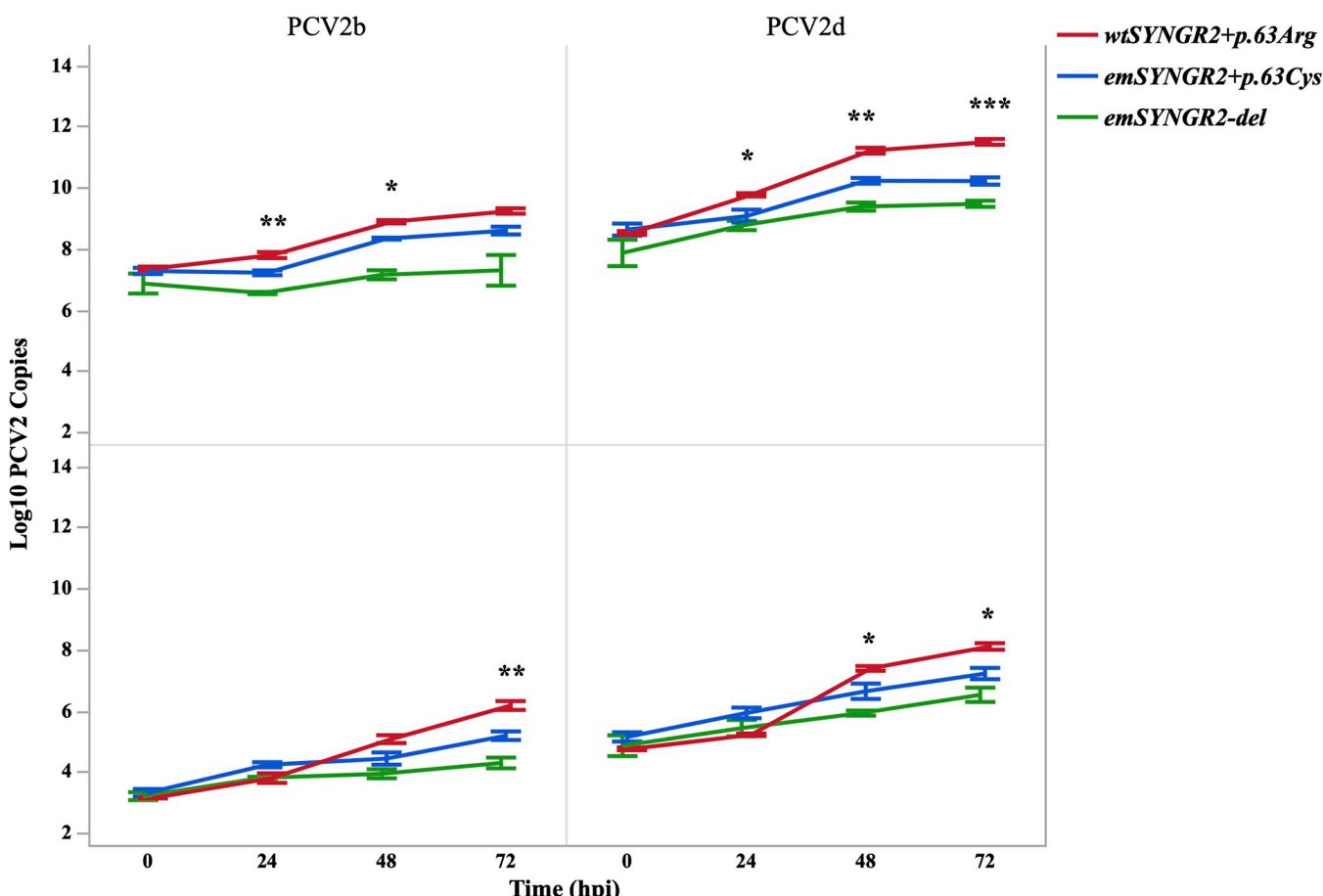

**Fig 1. PCV2 genome copy number in wildtype and edited PK15 clones following *in vitro* infection.** Log10 transformed viral genome copy number per well in the cell (top) and per uL in the supernatant (bottom) fractions across timepoints post infection with PCV2b or PCV2d (MOI = 0.00025). Error bars represent one standard error from the mean for three independent infection replicates. *P<0.05, **P<0.01, ***P<0.001.

PCV2a strains [10]. However, the PCV2b and PCV2d isolates differed within the Capsid tail region, with the PCV2d Capsid possessing two additional amino acids at the C-terminus, 234 (S) and 235 (E), predicted to further enhance host-cell receptor binding [10].

Following *in vitro* infection with the PCV2d isolate, the *emSYNGR2*$^{-del}$ cell line exhibited fewer PCV2d copies than either the *wtSYNGR*$^{+p.63Arg}$ and *emSYNGR2*$^{+p.63Cys}$ clones (P<0.05–0.001; Fig 1). A reduction in viral replication was also observed for the *emSYNGR2*$^{+p.63Cys}$ clone compared to *wtSYNGR2*$^{+p.63Arg}$ beginning at 24 hpi in the cell fraction and 48 hpi in the supernatant fraction (P<0.05; Fig 1). After 72 hpi, the amount of PCV2d copies increased 1,069-fold on average in *wtSYNGR2*$^{+p.63Arg}$ cells and 2,368-fold in the supernatant compared to only 39-fold in *emSYNGR2*$^{+p.63Cys}$ cells and 133-fold in supernatant (P<0.05).

## Effect of *SYNGR2 p.Arg63Cys* alleles on PCV2 infection are not due to differences in gene expression

Previous work revealed no differences in *SYNGR2* expression following *in vitro* infection of wildtype PK15 cells with PCV2b [5]. However, PCV2 has been shown to infect only a small proportion of cells in culture, which may have hindered detection of potential gene expression differences during infection. In attempt to enhance our ability to detect alterations in gene

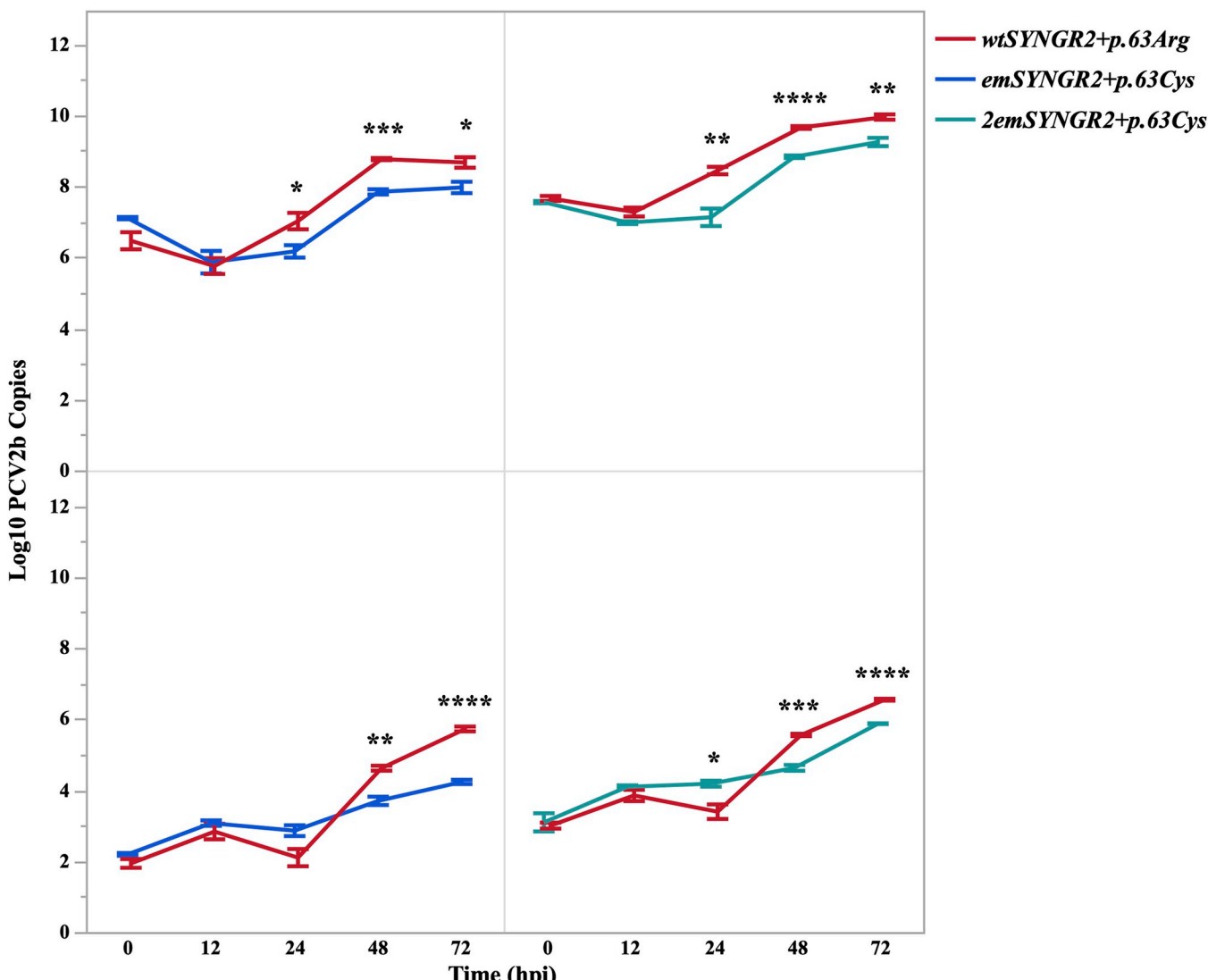

**Fig 2. PCV2b genome copy number in *wtSYNGR2*$^{+p.63Arg}$ and *emSYNGR2*$^{+p.63Cys}$ PK15 clones following *in vitro* infection.** Log10 transformed viral genome copy number per well in the cell (top) and per uL in the supernatant (bottom) fractions across timepoints post infection with PCV2b (MOI = 0.00075). Error bars represent one standard error from the mean for three independent infection replicates. *P<0.05, **P<0.01, ***P<0.001, ***P<0.0001.

expression, the *wtSYNGR2*$^{+p.63Arg}$ and *emSYNGR2*$^{+p.63Cys}$ cell lines were inoculated with the PCV2b isolate at a higher multiplicity of infection (MOI) as well as evaluated at an additional timepoint post infection (12 hpi) to better characterize the early stages of infection. Supporting these prior results, we observed no differences in *SYNGR2* expression levels between control and infected cells for either cell line (Fig C in S1 Text). However, *SYNGR2* expression was lower in the *emSYNGR2*$^{+p.63Cys}$ cells compared to *wtSYNGR2*$^{+p.63Arg}$ in both control and infected cells at various timepoints post infection (P<0.05; Fig C in S1 Text).

Consistent with the previous findings, a substantial reduction in the number of PCV2b copies was observed for *emSYNGR2*$^{+p.63Cys}$ compared to the *wtSYNGR2*$^{+p.63Arg}$ cell line starting at 24 hpi in the cell fraction (P<0.05) and 48 hpi in the supernatant (P<0.01; Fig 2). After 72 hpi, the amount of PCV2b copies increased 168-fold on average in *wtSYNGR2*$^{+p.63Arg}$ cells compared to only 8-fold in *emSYNGR2*$^{+p.63Cys}$ cells (P<0.05). This fold change was even more

pronounced in the supernatant fraction, with a 6,195-fold increase on average for $wtSYNGR2^{+-p.63Arg}$ compared to only 109-fold for $emSYNGR2^{+p.63Cys}$ (P<0.005). Comparing the slope of the line in the cell fraction between timepoints post infection, the rate of viral replication from 12 to 24 hpi was dramatically lower in $emSYNGR2^{+p.63Cys}$ cells compared to $wtSYNGR2^{+p.63Arg}$ (P<0.05). However, this difference was not maintained at later timepoints, with similar viral replication rates between cell lines after 24 hours. The effect of the early difference in cells is reflected in the supernatant fraction at subsequent timepoints, with substantially lower replication rates for $emSYNGR2^{+p.63Cys}$ relative to $wtSYNGR2^{+p.63Arg}$ after 24 hours (P<0.005).

Sequence comparison between the $emSYNGR2^{+p.63Cys}$ and $wtSYNGR2^{+p.63Arg}$ cell lines had previously confirmed no differences in the *SYNGR2* coding sequence besides the intended *63Arg* (C) > *63Cys* (T) allelic substitution (Fig B in S1 Text). However, additional genomic sequencing of non-coding regions revealed two intronic SNPs and one within the distal 3'UTR that differed between the two cell lines. Therefore, another edited PK15 clone homozygous for the *SYNGR2 p.63Cys* allele ($2emSYNGR2^{+p.63Cys}$), but otherwise identical to the $wtSYNGR2^{+-p.63Arg}$ cell line, was similarly infected with PCV2b to further validate the effect of the *SYNGR2 p.Arg63Cys* variant. Consistent with the first clone, a significant decrease in viral replication was observed in the $2emSYNGR2^{+p.63Cys}$ cell and supernatant fractions starting at 24 hpi compared to the $wtSYNGR2^{+p.63Arg}$ cell line (P<0.05) along with lower overall fold-change (P<0.01) and replication rate from 12 to 24 hpi in the cell fraction (P<0.05; Fig 2). No differences in gene expression were observed between control and infected $2emSYNGR2^{+p.63Cys}$ and $wtSYNGR2^{+p.63Arg}$ cells across timepoints (Fig C in S1 Text).

## Haplotype distribution across *Sus scrofa* subgroups provides insight into the evolutionary origin and positive selection of the *SYNGR2 p.63Cys* allele in domestic swine

Previous analysis of several domestic populations revealed significant variation in the frequency of the favorable *SYNGR2 p.63Cys* allele between breeds [5], prompting further inquiry into both the evolutionary origin of this allele and selection within the *Suidae* lineage. Comparative analysis of sequences available within the European Nucleotide Archive (ENA) from wild or domestic *S. scrofa* (n = 731) and relative species within the *Sus* genus (n = 20) identified 19 SNPs within the *SYNGR2* coding sequence, with *SYNGR2 p.Arg63Cys* being the only missense variant (Table B in S1 Text). The favorable *SYNGR2 p.63Cys* allele was found exclusively within domestic swine and was more prevalent in European (43.1%) than Asian (8.1%) with the highest frequencies in Duroc (68.3%) and Pietrain (41.2%) breeds (Table C and D in S1 Text). Genotypes for 15 SNPs (MAF>0.01) were used to assign haplotypes independently within three groups: *S. scrofa* domestic, *S. scrofa* wild boar, and *Sus* relatives. While no overlap was observed between *S. scrofa* (n = 10) and *Sus* relative haplotypes (n = 4), substantial overlap was present between domestic and wild boar with only two haplotypes, *Hap1* and *Hap8*, exclusively found in domestic swine (Table 1).

*Hap1* was the most prevalent haplotype within the domestic group (35%) and the only haplotype that included the favorable *SYNGR2 p.63Cys* allele. The second most frequent haplotype within the domestic group (*Hap2*, 30.9%) and most prevalent haplotype observed in wild boar (47.2%) was identical with *Hap1* except for the *SYNGR2 p.63Arg* allele. Both *Hap1* (*p.63Cys*) and *Hap2* (*p.63Arg*) were more predominant amongst European domestic (EUD) than Asian domestic (ASD) sequences, accounting for 77.6% of the haplotypes within EUD compared to only 17.7% in ASD (Fig D and Table E in S1 Text). This trend was even more dramatic across wild boar sequences, with *Hap2* being nearly fixed (97.7%) in European wild boar (EUW) and completely absent from Asian wild boar (ASW; Fig D and Table E in S1 Text). The geographic

**Table 1. *SYNGR2* haplotypes identified across *Suidae* groups.** (*187 = *SYNGR2* p.Arg63Cys; DM = Domestic, WB = Wild Boar, SR = *Sus* Relatives).

| *SYNGR2* Haplotype | SNP | | | | | | | | | | | | | | | Frequency | | | |
|---|---|---|---|---|---|---|---|---|---|---|---|---|---|---|---|---|---|---|---|
| | 105 | *187 | 195 | 198 | 210 | 321 | 324 | 465 | 516 | 540 | 576 | 588 | 597 | 603 | 645 | DM (n = 676) | WB (n = 55) | SR (n = 20) | All (n = 751) |
| Hap1 | G | T | C | C | T | T | C | C | C | C | C | G | C | C | C | 0.35 | – | – | 0.316 |
| Hap2 | - | C | - | - | - | - | - | - | - | - | - | - | - | - | - | 0.309 | 0.472 | – | 0.313 |
| Hap3 | - | C | - | T | - | C | - | - | - | T | - | - | T | T | - | 0.134 | 0.082 | – | 0.127 |
| Hap4 | - | C | - | T | - | C | - | T | T | T | - | - | - | - | - | 0.045 | 0.069 | – | 0.045 |
| Hap5 | - | C | - | T | - | C | - | - | - | - | - | - | - | - | - | 0.031 | 0.05 | – | 0.03 |
| Hap6 | - | C | - | - | - | C | - | - | - | - | - | - | - | - | - | 0.017 | 0.06 | – | 0.022 |
| Hap7 | - | C | - | - | C | C | - | - | - | - | - | - | - | - | - | 0.015 | – | – | 0.014 |
| Hap8 | - | C | - | T | - | C | - | - | - | T | - | - | - | - | - | 0.013 | – | – | 0.011 |
| Hap9 | - | C | T | T | - | C | - | - | - | - | - | - | - | - | - | 0.011 | 0.098 | – | 0.018 |
| Hap10 | - | C | - | T | - | C | - | - | T | T | - | - | - | - | - | – | 0.019 | – | 0.008 |
| Hap11 | - | C | - | - | C | - | - | - | - | - | T | A | - | - | T | – | – | 0.5 | 0.012 |
| Hap12 | - | C | - | - | C | - | T | - | - | - | T | A | - | - | T | – | – | 0.125 | 0.003 |
| Hap13 | - | C | - | - | C | C | - | T | - | T | - | - | T | - | - | – | – | 0.125 | 0.003 |
| Hap14 | A | C | - | - | C | - | - | - | - | - | T | A | - | - | T | – | – | 0.1 | 0.003 |
| | | | | | | | | | | | | | | | | 0.925 | 0.85 | 0.85 | 0.925 |

separation between European predominant (*Hap1* and *Hap2*) and Asian predominant (*Hap3-Hap10*) haplotypes is clearly illustrated in the haplotype network as well as the likely derivation of *Hap1* from the ancestral European *Hap2* background (Fig 3). Although present in the ASDO (ASD Other) group at a frequency of ~3%, *Hap1* was otherwise only present in the Meishan breed with a frequency of 17.3% (Fig E in S1 Text). However, *Hap1* was present in all seven EUD breeds, with the highest frequency in Duroc (67.8%) followed by Pietrain (43%; Fig E in S1 Text). EUD breeds with noted historical Asian introgression, such as Landrace and Large White/Yorkshire, exhibited greater haplotype diversity than other EUD breeds (Fig E in S1 Text). Specifically, compared to Landrace (50.8%) and Large White/Yorkshire (70.9–72.5%), *Hap1* and *Hap2* account for 100% of the haplotypes in Iberian, which is one of the few breeds with no written or molecular evidence of Asian introgression [11,12]. Together, these findings indicate that the favorable *SYNGR2* p.63Cys allele arose post-domestication in ancestral European domestic pigs from which modern European breeds were derived and was subsequently introduced into Asian domestic swine via European introgression [13].

To test for evidence of recent positive selection favoring the *SYNGR2* p.63Cys allele in domestic swine, the level of genetic diversity within the *SYNGR2* locus was compared between sequence groups homozygous for alternate *SYNGR2* p.Arg63Cys alleles (*Arg/Arg* = 356, *Cys/Cys* = 173). Based on 10 *SYNGR2* SNPs, the *Cys/Cys* group exhibited dramatically lower variation compared to the *Arg/Arg* group. Specifically, the observed heterozygosity within the *Cys/Cys* group was only 0.018 compared to 0.177 in the *Arg/Arg* group (Table 2). Heterogeneity was also observed across all *SYNGR2* SNPs within the *Arg/Arg* group, with eight out of 10 SNP exhibiting minor allele frequencies (MAF) greater than 0.05 (Table 3). In contrast, none of the SNPs within the *Cys/Cys* group had MAF greater than 0.05, with two being completely fixed (Table 3). In fact, the SNP with the highest MAF (0.02) in the *Cys/Cys* group exhibited a similar level of heterogeneity as the SNP with the lowest MAF (0.018) in the *Arg/Arg* group (Table 3). Together, these findings emphasize a significant reduction in genetic diversity across the *SYNGR2* locus in the derived *Cys/Cys* sequences relative to the ancestral *Arg/Arg* sequences, consistent with a rapid increase in frequency of the *SYNGR2* p.63Cys allele via positive selection in domestic swine.

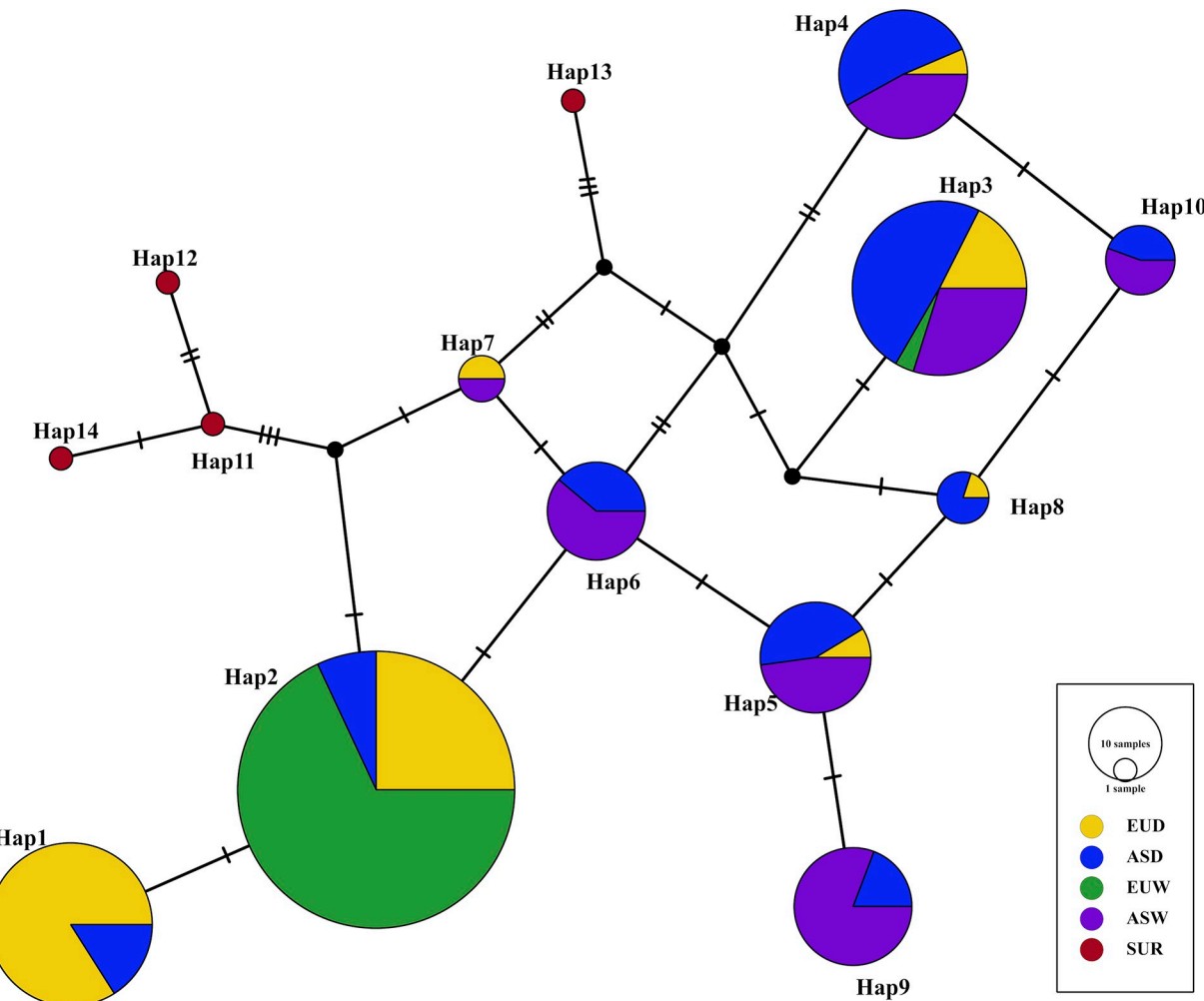

**Fig 3. Integer-joining network of *SYNGR2* haplotypes within *Suidae*.** Haplotype frequencies within each *S. scrofa* subgroup were applied to hypothetical sample sets of equal size. Node size represents haplotype frequency across *S. scrofa*, and different colored segments represent the proportion within each subgroup: European Domestic (EUD), European Wild Boar (EUW), Asian Domestic (ASD), Asian Wild Boar (ASW), *Sus* relatives (SUR). Black nodes represent inferred intermediate haplotypes not present in any samples within the dataset. Single nucleotide changes are represented as hatch marks.

## Signatures of adaptive evolution in the first intraluminal loop of the *SYNGR2* protein in mammals

Host VIPs often interact with even distantly related viruses and evolve to limit viral replication or pathogenesis [2]. To test whether this host-virus coevolutionary dynamic has left signatures of positive selection across the *SYNGR2* locus, we used the PAML package to fit models of molecular evolution to three taxonomic groups: mammals, primates, and even-toed ungulates

**Table 2. Parameters of genetic diversity across domestic groups with alternate homozygous *SYNGR2* p.Arg63Cys genotypes.** Average number of alleles (Num), effective number of alleles (Eff_num), observed heterozygosity (Ho), expected heterozygosity (Hs), and inbreeding coefficient (Gis) were estimated for each group.

| Group | Num | Eff_num | Ho | Hs | Gis |
|---|---|---|---|---|---|
| *Arg/Arg* | 2.1 | 1.562 | 0.177 | 0.316 | 0.439 |
| *Cys/Cys* | 1.8 | 1.018 | 0.018 | 0.018 | -0.013 |

**Table 3. Allelic frequencies for *SYNGR2* SNPs between domestic groups with alternate homozygous *SYNGR2 p.Arg63Cys* genotypes.** Bottom portion of table denotes the average (Avg), highest (Max), and lowest (Min) minor allele frequency (MAF) in each group.

| SNP | Allele | *Arg/Arg* | *Cys/Cys* | Overall |
|---|---|---|---|---|
| 195 | C | 0.982 | 1 | 0.988 |
| | T | 0.018 | 0 | 0.012 |
| 198 | C | 0.559 | 0.997 | 0.702 |
| | T | 0.441 | 0.003 | 0.298 |
| 210 | C | 0.031 | 0 | 0.021 |
| | T | 0.969 | 1 | 0.979 |
| 261 | C | 0.381 | 0.012 | 0.26 |
| | G | 0.579 | 0.988 | 0.713 |
| | T | 0.04 | 0 | 0.027 |
| 321 | C | 0.48 | 0.02 | 0.331 |
| | T | 0.52 | 0.98 | 0.669 |
| 465 | C | 0.921 | 0.997 | 0.946 |
| | T | 0.079 | 0.003 | 0.054 |
| 516 | C | 0.901 | 0.997 | 0.932 |
| | T | 0.099 | 0.003 | 0.068 |
| 540 | C | 0.644 | 0.98 | 0.754 |
| | T | 0.356 | 0.02 | 0.246 |
| 597 | C | 0.74 | 0.985 | 0.82 |
| | T | 0.26 | 0.015 | 0.18 |
| 603 | C | 0.744 | 0.985 | 0.823 |
| | T | 0.256 | 0.015 | 0.177 |
| **MAF** | Avg | 0.206 | 0.008 | 0.141 |
| | Max | 0.48 | 0.02 | 0.331 |
| | Min | 0.018 | 0 | 0.012 |

[14,15]. Overall $d_N/d_S$ ($\omega$) estimates indicated that negative selection is acting to conserve the *SYNGR2* protein sequence across mammalian species ($\omega$ = 0.09–0.183; Table 4). However, comparison between site-specific models supports positive selection on a subset of codons located exclusively within the two intraluminal loop domains in each taxonomic group (P<0.01; Table 4). Residue 53, which is located within the first intraluminal loop and only 10 residues upstream from the *SYNGR2 p.Arg63Cys* site, was identified in all three groups (Fig 4). Two other residues within the first intraluminal loop, 55 and 57, and two residues within the second intraluminal loop, 135 and 139, were also identified in the mammalian and/or even-toed ungulate groups (Fig 4). These findings highlight the intraluminal loops as sites of potential adaptive evolution to environmental cues, including host-pathogen interactions.

## Discussion

### Proposed mechanism of *SYNGR2* function during early stages of PCV2 infection

Previously, our group uncovered a missense variant (*SYNGR2 p.Arg63Cys*) within the host Synaptogyrin-2 (*SYNGR2*) gene statistically associated with PCV2b viral load and immune response following *in vivo* experimental challenge [5]. *In vitro* infection of a predicted *SYNGR2* knock-out edited PK15 clone with the same PCV2b isolate demonstrated clear involvement of *SYNGR2* in viral replication [5]. However, direct experimental evidence for the

**Table 4. Analysis of positive selection within *SYNGR2* across mammalian species using site-specific models of evolution.** For each model, the dN/dS ratio was estimated for the entire gene (dN/dS) and across each site within *SYNGR2* using the corresponding gene tree. A significant likelihood ratio test (LRT) statistic indicates the model allowing positive selection (M8, $\omega \leq 1$ or $\omega > 1$) is a better fit than the null model (M7, $\omega \leq 1$). The number of positive selection sites (PSS) is reported for each cluster.

| | Mammals | | | Primates | | | Even-Toed Ungulates | | |
|---|---|---|---|---|---|---|---|---|---|
| **Model** | **dN/dS** | **PSS ($\omega>1$)** | **LRT** | **dN/dS** | **PSS ($\omega>1$)** | **LRT** | **dN/dS** | **PSS ($\omega>1$)** | **LRT** |
| **M7** | 0.183 | -- | 65.79 (P<0.001) | 0.090 | – | 13.41 (P<0.01) | 0.142 | -- | 27.14 (P<0.001) |
| **M8** | 0.167 | 4 (3%) | | 0.102 | 1 (0.6%) | | 0.192 | 4 (3.6%) | |

causality of the *SYNGR2 p.Arg63Cys* variant or consistency of its effect across PCV2 subtypes had yet to be provided. In this study, we validated the function of the *SYNGR2 p.Arg63Cys* variant in PCV2 infection using multiple *in vitro* models and PCV2 isolates. Specifically, CRISPR-Cas9 mediated allelic substitution for the *SYNGR2 p.63Cys* allele in PK15 cells resulted in significant reductions in viral replication consistent across PCV2 subtypes and inoculate concentration. These findings demonstrate a direct effect of the host *SYNGR2* gene and causality of the associated *SYNGR2 p.Arg63Cys* substitution on PCV2 replication. Additionally, the consistent effect of *SYNGR2* genotype on both PCV2b and PCV2d subtypes despite distinct infection kinetics, suggests that PCV2 evolution has yet to successfully counter the protection provided by the *SYNGR2 p.63Cys* allele.

While further analysis is necessary to understand the exact functional mechanism, the lag in PCV2 replication during the first 24 hours post infection in edited clones compared to wild-type cells, highlights a potential function of *SYNGR2* in the early stages of PCV2 infection. This timeframe corresponds with the first intracytoplasmic phase of PCV2 morphogenesis, in

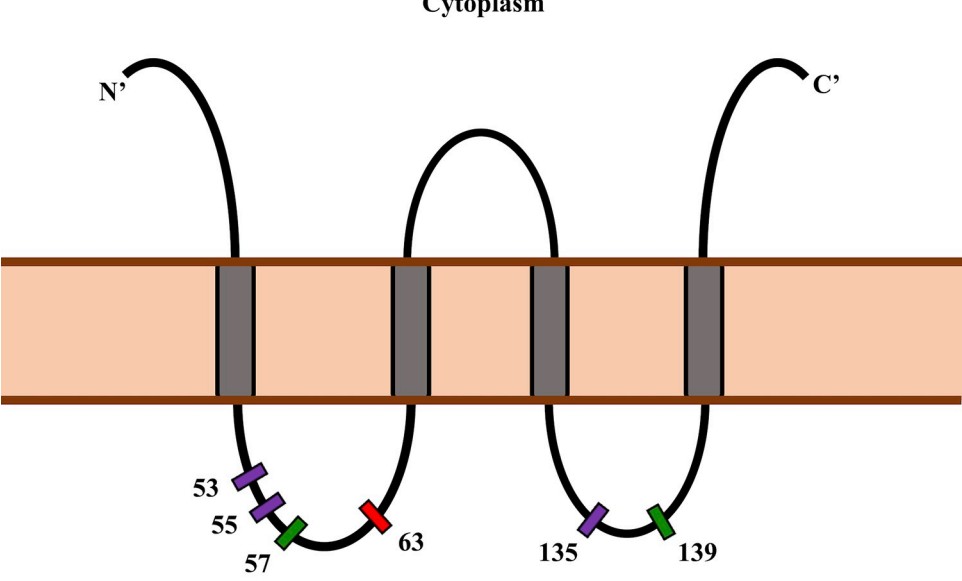

**Fig 4. Position of putative positive selection sites identified across mammals within the *SYNGR2* protein.** Schematic depicting the four transmembrane domains (gray boxes), two intraluminal loops, one cytoplasmic loop domain and cytoplasmic N- and C-termini. Amino acid residues identified as positive selection sites within one (green) or more (purple) taxonomic groups are represented relative to the *SYNGR2 p.Arg63Cys* polymorphism (red).

which virions are internalized via endocytosis into early endosomes and subsequent transloca-tion to the nucleus for viral replication [16,17]. *SYNGR2* has been identified as a transmem-brane component of cytoplasmic vesicles as well as involved in vesicle biogenesis found to be necessary for the promotion of SFTSV replication, a human RNA virus [6,7]. Additionally, *SYNGR2* has been further characterized as a crucial factor for internalization of the *A. actino-mycetemcomitans* cytotoxin active subunit (CdtB) via clathrin-mediated endocytosis [8,9]. Therefore, a function of *SYNGR2* predominantly in the early stages of PCV2b infection is a logical conjecture. We propose that the detrimental effects on PCV2 replication resulting from *SYNGR2* knock-out or the *SYNGR2 p.63Cys* allele is due to impaired endocytosis and/or slower translocation of virions to the nucleus for replication.

### Origin and positive selection of the *SYNGR2 p.63Cys* allele within domestic swine

To investigate the evolutionary origin and history of the favorable *SYNGR2 p.63Cys* allele within the *Suidae* lineage, we evaluated *SYNGR2* sequence data from 751 wild and domestic pigs available within the European Nucleotide Archive database. In this dataset, the *SYNGR2 p.63Cys* allele was found to be unique to domestic pigs and more prevalent in European than Asian breeds, with the highest frequencies in Duroc and Pietrain breeds. Additionally, a single *SYNGR2* haplotype (*Hap1*) was defined by the *SYNGR2 p.63Cys* substitution and was likely derived form an ancestral haplotype (*Hap2*) nearly fixed within European wild boar and completely absent from Asian wild boar. A striking difference was also observed between European and Asian breeds, with *Hap1* prevalent in all European compared to only one Asian breed (Meishan). Given the historical prominence of the Meishan breed, if the *SYNGR2 p.63Cys* allele had originated in this breed it would more than likely be present in many other Asian domestic breeds, which we did not observe. Additional insight is provided by the pres-ence of only *Hap1* and *Hap2* in Iberian samples. Coinciding with noted continuous introgres-sion between Iberian and European wild boar [12,18–20], *Hap2* was present at the highest frequency in Iberian (88.2%) compared to all other European breeds. However, Iberian pigs are also one of the few European breeds with no known written or molecular evidence of Asian introgression [11,12]. Therefore, we propose that the favorable *SYNGR2 p.63Cys* allele arose post-domestication in ancestral European domestic pigs from which modern European breeds were formed (Fig 5). While many studies emphasize introgression of Asian domestic into European domestic swine during the 18th century, a few studies have also noted possible European introgression into Asian populations, particularly within Northern China [13,21,22]. Therefore, the presence of the *SYNGR2 p.63Cys* allele in Asian domestic swine under this proposed model can be explained by subsequent introgression of European haplo-types (Fig 5).

Despite the seemingly recent emergence of this allele post-domestication, the extremely high frequency within Duroc and notable frequencies across European breeds suggests poten-tial positive selection within domestic swine. As previously stated by Walker et al. (2018), the Duroc breed is noted for lean growth and has historically exhibited strong selective pressure for this trait throughout the 20th century, coinciding with the earliest detection of PCV2 strains in archival samples from the 1960's [23]. Based on the effects that even mild PCV2 infections can have on growth, we postulated that the *SYNGR2 p.63Cys* allele may have been unknow-ingly selected for in this breed due to the relationship between PCV2 viral load and overall fit-ness [5]. A similar mechanism of selection, albeit of variable intensity, on other correlated production traits could also have led to the elevated frequency of this allele across European domestic breeds. To test this hypothesis, we compared the genetic diversity exhibited across 10

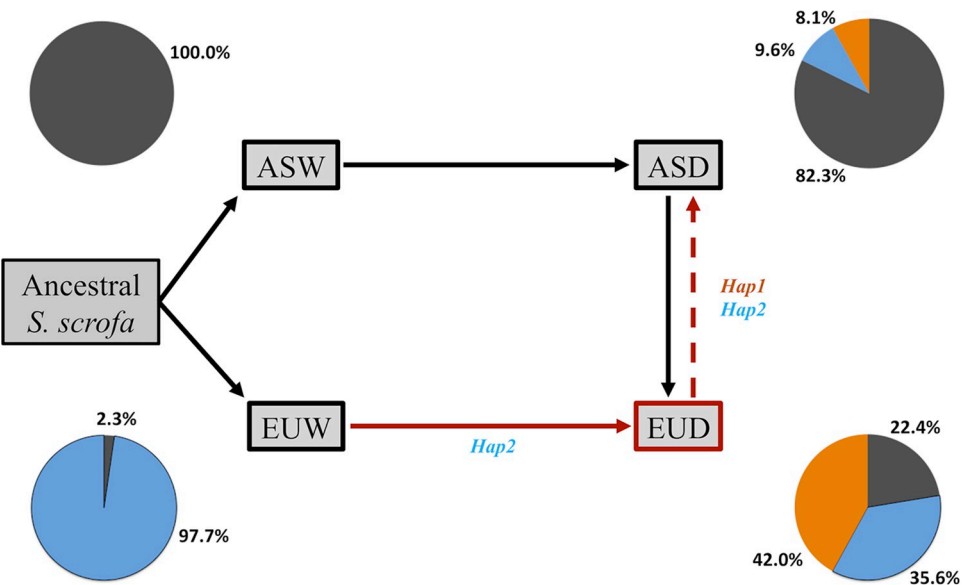

**Fig 5. Proposed scenario of *SYNGR2 p.63Cys/Hap1* origin in European domestic swine.** Solid lines reflect well-documented and experimentally supported directions of gene flow between populations throughout swine domestication. Dashed lines represent less documented directions of gene flow necessary for the proposed scenario of origin in EUD (red lines). Pie charts represent the frequency of *Hap1* (orange) and *Hap2* (blue) in each *S. scrofa* subgroup. (ASW = Asian Wild Boar, ASD = Asian Domestic, EUW = European Wild Boar, EUD = European Domestic).

*SYNGR2* SNP between domestic pigs with alternate homozygous *SYNGR2 p.Arg63Cys* genotypes. Overall, this analysis revealed a dramatic reduction in heterozygosity amongst *Cys/Cys* compared to *Arg/Arg* individuals, supporting recent positive selection favoring the *SYNGR2 p.63Cys* allele.

## A model of host-virus coevolution across mammals

Given that *SYNGR2* may represent a common mammalian VIP, we also assessed for signatures of adaptive evolution within *SYNGR2* across mammalian species, which would support a model of repeated host-virus coevolution. Corresponding with reports of strong purifying selection acting on non-immune VIPs [2], we observed an overall signature of negative selection for *SYNGR2* across mammals. However, we also found evidence of adaptive evolution with positively selected sites identified exclusively within the intraluminal loop domains and nearby the location of the *SYNGR2 p.Arg63Cys* substitution. Together, these findings indicate that *SYNGR2* may reflect a component of the host-virus evolutionary arms race common across mammals with *SYNGR2 p.Arg63Cys* representing a species-specific example of recent putative adaptive evolution against PCV2 infection.

## Materials and methods

### PK15 cell culture

The porcine kidney cell line (PK15) was grown in DMEM high glucose media supplemented with 10% FBS and 1% Penicillin-Streptomycin (5,000 U/mL) at 37° C with 5% $CO_2$. The cells were passaged every 3–4 days (1:10 split) and allowed to passage at least twice after thawing before being plated for transfection or infection protocols.

## PCV2 isolates

The PCV2b strain (UNL2014001, accession KP016747.1) is the same isolate used in prior *in vivo* and *in vitro* infections [5] and was previously propagated in swine testicular (ST) cells with a final titer of $1x10^4$ 50% tissue culture infection dose ($TCID_{50}$) as described by McKnite et al. (2014)[24]. The PCV2d isolate was obtained from the Kansas State University (KSU) diagnostic laboratory from an animal of unknown clinical status. The PCV2d strain was propagated in PK15 cells and harvested via three consecutive freeze-thaw cycles. To determine $TCID_{50}$, the harvested virus was titrated in PK15 cells and analyzed via immunocytochemistry after 72 hours. The cells were incubated with a polyclonal anti-PCV2 primary antibody (ISU-3, 1:100) followed by a donkey anti-rabbit IgG secondary antibody conjugated with horse-radish peroxidase (AB_2534697, Invitrogen, 1:2,000). PCV2d-infected cells were stained using the peroxidase AEC substrate kit (Vector Laboratories) and visualized via light microscopy. The final PCV2d titer was estimated to be ~$1x10^{5.5}$ $TCID_{50}$. The complete PCV2d genome was sequenced via dideoxy sequencing and compared with the complete PCV2b genome sequence to assess overall similarity. The percent identity between the two isolates for both the Replicase and Capsid viral proteins was also assessed for the corresponding nucleotide and predicted amino acid sequences.

## *In vitro* editing of *SYNGR2* in PK15 cells

The knock-out *SYNGR2* clone (*emSYNGR2^-del^*) utilized in this study is the same clone (*E1*) previously generated by Walker et al. (2018)[5]. To generate PK15 clones homozygous for the alternate *SYNGR2 p.63Cys* allele, a single guide RNA was utilized targeting a protospacer adjacent motif (PAM) recognized by the Cas9 enzyme that overlapped the *SYNGR2 p.Arg63Cys* locus. The guide RNA was hybridized with fluorescently labeled Alt-R CRISPR-Cas9 tracrRNA ATTO 550 (IDT) and Alt-R S.p. Hifi Cas9 Nuclease V3 (IDT) following the manufacturer's protocol to form Ribonucleoprotein (RNP) complexes. These RNP complexes were reverse transfected into PK15 cells in 12-well plates ($4.0x10^5$ cells/well) using Lipofectamine RNAi-MAX transfection reagent (Invitrogen) at a final concentration of 10nM. Cutting efficiency of this guide RNA was evaluated as previously described by Walker et al. (2018) using the T7E1 mutation detection kit (NEB). An 80 bp single stranded DNA template (IDT) homologous to the non-targeting strand and encoding the desired *SYNGR2 p.63Cys* allele was reverse transfected along with RNP complexes into PK15 cells. After 24 hours post transfection, the cells were collected and sorted using Fluorescence Activated Cell Sorting into 96 well plates to generate single-cell clones. Genomic DNA from each single-cell clone was extracted using Quick-Extract DNA extraction solution following the manufacturer's protocol (Lucigen). Each clone was genotyped for the *SYNGR2 p.Arg63Cys* variant by KASPar assay (LGC). RNA was extracted from selected clones using the RNeasy mini kit (Qiagen) and complementary DNA (cDNA) was generated using the First Strand cDNA Synthesis Kit (Cytiva). The *SYNGR2* locus was sequenced via dideoxy sequencing at both the genomic and transcriptomic levels to confirm clone genotypes and assess for potential off-target effects. Sequences were aligned via clustal omega (https://www.ebi.ac.uk/Tools/msa/clustalo) and annotated using Jalview 2.11.2.7 software [25].

## *In vitro* infection of PK15 cells

Wildtype PK15 and edited clones were cultured in 12-well plates (4 cm$^2$) with $5.0x10^5$ cells per well and infected when 80–100% confluent with either the PCV2b or PCV2d isolates. Infections were conducted a total of three times, each at a different time and cell passage, to obtain independent infection replicates. Cells were inoculated with PCV2d at the same multiplicity of

infection (MOI) = 0.00025 used in the previous study [5] while a higher MOI = 0.00075 was used for PCV2b in attempt to better detect differences in gene expression due to infection. The inoculate was removed after a one-hour incubation period and cells were washed twice with PBS followed by supplementation with fresh media (DMEM high glucose with 2% FBS). The cells were incubated at 37° C with 5% $CO_2$ for up to 5 days. Control cells were inoculated with plain DMEM high glucose media and maintained the same way as infected cells. Supernatant and cell samples were collected from corresponding wells at specific time points post infection and frozen at -80° C. PCV2d viral DNA was extracted from pelleted cell and supernatant (200 uL) samples using the QIAamp Blood DNA mini kit (Qiagen). PCV2b viral DNA and host cell RNA were simultaneously extracted from cell pellets using the AllPrep DNA/RNA mini kit (Qiagen). cDNA was generated from host-cell RNA samples as described above.

## PCV2 genome copy number and host *SYNGR2* expression profiling

Expression of *SYNGR2* and PCV2 viral genome copy number were quantified across time points post infection using TaqMan Master Mix and the CFX384 Real Time PCR system (BioRad). The qPCR assays were designed using the IDT Realtime PCR Tool software (www. idt.com). A conserved viral genomic sequence within ORF2 (Capsid) was targeted to enable quantification of PCV2b and PCV2d with the same assay. PCV2 genome copy number per microliter for each viral DNA sample (50 uL) was quantified by plotting cycle crossing thresholds (CT) obtained for the technical triplicates on a standard curve generated by serial dilution of a PCR-amplified PCV2b control. The PCV2 copy number was then adjusted to reflect the total amount per well ($\frac{copies}{uL} \times 50\ uL$) in the cell fraction and per microliter ($\frac{copies}{uL} \div 4\ dilution\ factor$) in the supernatant fraction. The PCV2 copy number across timepoints post infection were log10 transformed and compared between PK15 clones by Tukey's honest significance test or t-test. Host cell cDNA samples were used to profile expression of *SYNGR2* in wildtype and edited PK15 clones. The qPCR assay targeted the 3'UTR of the host SYNGR2 transcript. Simultaneous profiling of the ribosomal protein L32 (*RPL32*) gene was used for normalization. Mean normalized expression (MNE) values were calculated based on CT values obtained for the technical triplicates taking qPCR efficiencies into account [26]. MNE values for wildtype (*wtSYNGR2*$^{+p.63Arg}$) and edited (*emSYNGR2*$^{+p.63Cys}$) PK15 clones were log10 transformed and compared by t-test.

## Collection, alignment, and variant discovery of *Suidae SYNGR2* sequences

DNA sequences generated from individual animals with clearly defined wild or domestic purebred status were obtained from public sources as described in the Supplementary information (S1 Text). The sequencing reads for each sample were individually aligned to the coding region of the reference *Sus scrofa SYNGR2* transcript sequence (XM_021066554.1, 85-759bp) using Standard Nucleotide BLAST software (NCBI). Aligned reads were visualized using the multi-sequence alignment viewer and deviations from the reference *SYNGR2* sequence were recorded for each sample. Genotypes were called based on the nucleotides present across aligned reads corresponding to each variant site. To minimize inclusion of false variants due to sequencing or alignments errors, focus was placed on variants identified across multiple samples within the dataset.

## Haplotype analysis

*SYNGR2* haplotypes were identified using Haploview 4.2 software [27]. To account for differences in sample size, predominant haplotypes were identified separately for each of the three

broad groups (Domestic, Wild Boar, and *Sus* Relatives). Only biallelic variants with minor allele frequencies (MAF) above 0.01 within at least one group were included in the haplotype analyses. Haplotype frequency (HAF) thresholds were independently set for each group according to sample size: *Sus* relatives (HAF>0.049), Wild Boar (HAF>0.018), and Domestic (HAF>0.01). The haplotypes were compared across groups to compile a set of *SYNGR2* haplotypes. A haplotype network based on *SYNGR2* coding sequences corresponding to each haplotype was generated using POPART software [28]. Specifically, the Integer Joining Network (IJN) method was utilized, which generates a network based on an inferred neighbor-joining tree and allows for the inclusion of nodes that reflect predicted ancestry and reduce edge length (reticulation tolerance = 0.5). To account for differences in sample size among groups, haplotype frequencies based on the actual dataset were applied to hypothetical groups of equal size (n = 100). The node size reflects the haplotype frequency across groups and the pie segments within each node represent the number of samples with that haplotype in each group.

### Selection analysis within domestic swine

A total of 529 domestic samples with homozygous *SYNGR2 p.Arg63Cys* genotypes were divided into two groups, *Arg/Arg* (n = 356) and *Cys/Cys* (n = 173), regardless of breed or geographic origin. Genotypes for *SYNGR2* SNPs with MAF>0.01 (excluding *SYNGR2 p. Arg63Cys*) were included in subsequent analyses. Allelic frequencies and parameters of genetic diversity were estimated using GenoDive 3.5 software [29]. Allelic frequency was calculated for each SNP both overall and within genotype groups. The genetic diversity tool was used to evaluate average number of alleles or allelic richness (num), effective number of alleles (Eff_num), observed heterozygosity (Ho), expected heterozygosity (Hs), total heterozygosity (Ht), and the inbreeding coefficient (Gis) for each genotype group.

### *SYNGR2* sequence and selection analyses across mammalian taxonomic groups

Reference transcript sequences encoding the common *SYNGR2* isoform (224 aa) were obtained from NCBI for a total of 40 mammalian species (Table A in S1 Text). Each sequence was translated into the corresponding predicted protein sequence using an online DNA to protein translation software (http://bio.lundberg.gu.se/edu/translat.html). The sequences were categorized into three taxonomic groups: Mammals (n = 22), Primates (n = 16), and Even-Toed Ungulates (n = 10). The DNA and protein sequences within each group were aligned using Clustal Omega with codon specific alignment (PAL2NAL) as outlined in Jeffares et al. (2015)[14]. The amino acid alignments were used to estimate the phylogenetic tree or "gene" tree via the RAxML tool with parameters outlined in Jeffares et al. (2015)[14].

We used CODEML to implement a maximum likelihood approach within the PAML package [15] that fits aligned sequences to specified models of evolution by estimating the ratio of non-synonymous ($d_N$) to synonymous ($d_S$) substitutions ($d_N/d_S$ or $\omega$). Specifically, we analyzed each taxonomic group of aligned *SYNGR2* sequences using site-specific models of evolution, in which $\omega$ is estimated for each site (i.e. codon) within the coding sequence. To test for evidence of positive selection within *SYNGR2*, two different models were run for the same data set representing the null hypothesis (M7), which does not allow for positively selected sites ($\omega \leq 1$), and the alternate hypothesis (M8), which does allow for positively selected sites ($\omega \leq 1$, $\omega > 1$). The CODEML models were run as described in Jeffares et al. (2015)[14]. The likelihood ratio test (LRT) values for the two models were then compared via chi-square distribution (df = 2) to determine which is the best fit for the data [14]. Under the M8 model, any sites within the protein sequence classified as $\omega > 1$ are specified in the output file and represent

putative sites of positive selection [14]. The selection analysis was conducted twice for each group using either the gene or species tree to account for potential differences in evolutionary rates at the gene and species level. We observed a high level of concordance between the two analyses for each group so only the gene tree results are presented in this study.

## Supporting information

**S1 Text. Supporting figures and tables. Fig A. CRISPR-Cas9 guide RNA and template design for generation of clones homozygous for alternate *SYNGR2 p.63Cys* allele.** The selected guide RNA (sg_AF) targeted a PAM site within exon 2 (gray box) that included the *SYNGR2 p.63Arg* allele (red). An 80 bp ssDNA sequence (thick black line) homologous to the non-targeting strand and encoding the alternate *SYNGR2 p.63Cys* allele (yellow) was used as a template for homology-directed repair. **Fig B. Alignment of the *SYNGR2* coding sequence for wildtype and edited PK15 cell lines.** Sequences were obtained by targeted amplification of the *SYNGR2* transcript and Sanger sequencing. Position of the substitution of *p.63Arg* (C) to *p.63Cys* (T) in the edited PK15 clones, *emSYNGR2*$^{+p.63Cys}$ and *2emSYNGR2*$^{+p.63Cys}$, is highlighted in red. The 106 bp deletion in the predicted *SYNGR2* knock-out PK15 clone, *emSYNGR2*$^{-del}$, is represented by dashes. **Fig C. Expression of *SYNGR2* in wildtype and edited PK15 following PCV2b infection mock-infected control cells.** Expression represented as Log10 transformed mean normalized expression (MNE) across three independent replicates with error bars representing one standard error from the mean. Samples collected from control and infected cells across timepoints post PCV2b infection (MOI = 0.00075). Letters denote significant differences in gene expression between cell lines within treatment group (C = control, I = infected) or between treatment groups within cell lines (wt = wildtype, em = edited). *P<0.05, **P<0.01. **Fig D. Frequency of *SYNGR2* haplotypes across geographic and domestic/wild *S. scrofa* subgroups.** The two haplotypes that differ by only the *SYNGR2 p.Arg63Cys* allele, *Hap1* (*Cys*) and *Hap2* (*Arg*), are represented as independent pie segments. Hap3-Hap10 and rare haplotypes were combined into a single category denoted as "Other". (ASW = Asian Wild Boar, ASD = Asian Domestic, EUW = European Wild Boar, EUD = European Domestic). **Fig E. Frequency of *SYNGR2* haplotypes across domestic breeds.** The two haplotypes that differ by only the *SYNGR2 p.Arg63Cys* SNP, *Hap1* (*Cys*) and *Hap2* (*Arg*), are represented as independent pie segments. Hap3-Hap10 and rare haplotypes were combined into a single category denoted as "Other". (BR = Berkshire, DR = Duroc, IB = Iberian, PI = Pietrain, LR = Landrace, LW = Large White, YR = Yorkshire, EUDO = European Domestic Other, EH = Erhualian, MS = Meishan, ASDO = Asian Domestic Other). **Table A. Mammalian *SYNGR2* transcript sequences and taxonomic groups. Table B. *SYNGR2* SNP identified across *Suidae* sequences.** Each *SYNGR2* SNP is denoted by nucleotide position within the *SYNGR2* coding sequence. (*187 = *SYNGR2 p.Arg63Cys*; SSC12 = chromosome 12, *S.scrofa*11.1). **Table C. Allelic frequencies for *SYNGR2* SNP across *Sus scrofa* subgroups.** (*187 = *SYNGR2 p.Arg63Cys*; EUD = European Domestic, EUW = European Wild, ASD = Asian Domestic, ASW = Asian Wild). **Table D. Allelic frequencies for *SYNGR2* SNP across domestic breeds.** (*187 = *SYNGR2 p.Arg63Cys*; BR = Berkshire, DR = Duroc, IB = Iberian, PI = Pietrain, LR = Landrace, LW = LargeWhite, YR = Yorkshire, EUDO = European other, EH = Erhualian, MS = Meishan, ASDO = Asian other). **Table E. Frequency of *SYNGR2* haplotypes across *S. scrofa* subgroups.** (EUD = European Domestic, EUW = European Wild, ASD = Asian Domestic, ASW = Asian Wild).
(DOCX)

**S1 Table. SYNGR2 expression and viral titer data following in vitro PCV2 infection of PK15 cells.**
(XLSX)

## Author Contributions

**Conceptualization:** Hiep L. Vu, Kristi L. Montooth, Daniel C. Ciobanu.

**Formal analysis:** Lianna R. Walker.

**Funding acquisition:** Daniel C. Ciobanu.

**Investigation:** Lianna R. Walker, Hiep L. Vu, Daniel C. Ciobanu.

**Methodology:** Lianna R. Walker, Hiep L. Vu, Kristi L. Montooth, Daniel C. Ciobanu.

**Project administration:** Daniel C. Ciobanu.

**Resources:** Daniel C. Ciobanu.

**Software:** Kristi L. Montooth.

**Supervision:** Hiep L. Vu, Kristi L. Montooth, Daniel C. Ciobanu.

**Validation:** Lianna R. Walker.

**Visualization:** Lianna R. Walker.

**Writing – original draft:** Lianna R. Walker, Daniel C. Ciobanu.

**Writing – review & editing:** Hiep L. Vu, Kristi L. Montooth, Daniel C. Ciobanu.

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
