## [Decision Letter · Decision Letter 0]

3 Oct 2023

Dear Dr Ciobanu,

Thank you very much for submitting your Research Article entitled 'SYNGR2 and PCV2: A Model of Host Adaptive Evolution to Viral Pathogens' to PLOS Genetics.

The manuscript was fully evaluated at the editorial level and by independent peer reviewers. The reviewers appreciated the attention to an important problem, but raised some substantial concerns about the current manuscript. Based on the reviews, we will not be able to accept this version of the manuscript, but we would be willing to review a much-revised version. We cannot, of course, promise publication at that time.

If you decide to revise the manuscript for further consideration at PLOS Genetics, please aim to resubmit within the next 60 days, unless it will take extra time to address the concerns of the reviewers, in which case we would appreciate an expected resubmission date by email to plosgenetics@plos.org.

We are sorry that we cannot be more positive about your manuscript at this stage. Please do not hesitate to contact us if you have any concerns or questions.

Yours sincerely,

Martien Groenen, PhD

Academic Editor

PLOS Genetics

Gregory Barsh

Editor-in-Chief

PLOS Genetics

While the three reviewers find the results interesting for a broader scientific audience, reviewers 2 and 3 have a number of comments that first need to be addressed before the manuscript can be accepted for publication in PLoS genetics.

Reviewer's Responses to Questions

**Comments to the Authors:**

Reviewer #1: This is an interesting study that dissects the functional role of a missense substitution in host SYNGR2 gene ( SYNGR2 p.Arg63Cys ) in the replication of porcine circovirus 2. It nicely uses a variety of developed genetic tools to profile the anti-virus mechanism of SYNGR2 during early stages of PCV2 infention. The experiments are generally well executed, logical, and appropriately interpreted. Moreover, the authors use existing datasets and simulations provides some novel insights including origin and subsequent positive selection of the favorable SYNGR2 p.63Cys allele within European pig populations. The Discussion is good and addresses the relevant fields and implications of the reseach. The manuscript is clearly written and easy to follow. The results will be of interest to the broad audience of Plos Genetics.

Reviewer #2: The authors report the association of a variant of the synaptogyrin-2 transmembrane protein (SYNGR2 p.Arg63Cys) with the course of infection with procine circovirus subtypes. The authors have previously shown that SYNGR2 knockout cell lines PK15 in vitro and the host missense variant (p.Arg63Cys) in vivo are associated with reduced viral load. In the present study, the authors aim to show direct experimental evidence for the function of the SYNGR2 86 p.Arg63Cys variant in the early stages of infection using in vitro models. The authors further show that the variant originates in and is prevalent in European pig populations. This is an interesting study. Using CRISPR/Cas9 edited PK15 cell lines, a reduced viral load was shown for both SYNGR2 deleted (emSYNGR2-del) and SYNGR2 p.63Cys variant versus SYNGR2 p.63 Arg for PCV2b and PCV2d. Further, variation of SYNGR2 but not infectious status is relevant for difference in de expression of SYNGR2. Further investigation of haplotypes in different populations from freely available sequence data shows the prevalence of the beneficial haplotype in European pig breeds and a lower diversity over 10 SNPs in the SYNGR2 locus. The tag of the publication "A Model of Host Adaptive Evolution to Viral Pathogens" is a bit lurid and misleading and with the comparisons across species not core to the study. The manuscript should be rewritten accordingly.

Figure 1: "three independent replicates": does this mean replicates of qPCR reactions; 12-well plates were used for cell culture, so 12 replicates per clone (supernatant, cell pellet) can be expected; I assume it is 12 wells divided by 4 time points; this needs to be clearly shown.

Reviewer #3: Dear authors,

I have really enjoyed reviewing this original research manuscript. I have a few questions regarding the establishment of the cell edited models. For sure you have these results and I think it is important that these should be included even if as supplemental material since these are the basis for the whole manuscript. I missed the validation of the knock-out and of the editing. I therefore request that PCR validation, Western-blot and sanger sequencing validation would be added to the manuscript.

Then regarding the discussion in line 296, your discussion does not mention the fact that there is continuous genetic introgression of EWB into Iberian pigs, and recent studies have shown how some samples in ENA cluster closer with EWB and not with other ED. I believe that is more a cause for this result than the fact that it is not introgressed with Asian.

**Have all data underlying the figures and results presented in the manuscript been provided?**

Reviewer #1: Yes

Reviewer #2: Yes

Reviewer #3: Yes

PLOS authors have the option to publish the peer review history of their article (what does this mean?). If published, this will include your full peer review and any attached files.

Reviewer #1: No

Reviewer #2: No

Reviewer #3: **Yes: **Andreia J. Amaral

---

## [Editor Report · Decision Letter 1]

24 Oct 2023

Dear Dr Ciobanu,

We are pleased to inform you that your manuscript entitled "Functional and Evolutionary Analysis of Host Synaptogyrin-2 in Porcine Circovirus Type 2 Susceptibility" has been editorially accepted for publication in PLOS Genetics. Congratulations!

Yours sincerely,

Martien Groenen, PhD

Academic Editor

PLOS Genetics

Gregory Barsh

Editor-in-Chief

PLOS Genetics

**Data Deposition**

http://datadryad.org/submit?journalID=pgenetics&manu=PGENETICS-D-23-00923R1

**Press Queries**

---

## [Editor Report · Acceptance letter]

9 Nov 2023

PGENETICS-D-23-00923R1 

Functional and Evolutionary Analysis of Host Synaptogyrin-2 in Porcine Circovirus Type 2 Susceptibility 

Dear Dr Ciobanu, 

We are pleased to inform you that your manuscript entitled "Functional and Evolutionary Analysis of Host Synaptogyrin-2 in Porcine Circovirus Type 2 Susceptibility" has been formally accepted for publication in PLOS Genetics! Your manuscript is now with our production department and you will be notified of the publication date in due course.

With kind regards,

Zsofi Zombor

PLOS Genetics

On behalf of:
